# Correlation and Difference between Core Micro-Organisms and Volatile Compounds of Suan Rou from Six Regions of China

**DOI:** 10.3390/foods11172708

**Published:** 2022-09-05

**Authors:** Kuan Lu, Xueya Wang, Jing Wan, Ying Zhou, Hongying Li, Qiujin Zhu

**Affiliations:** 1Key Laboratory of Plant Resource Conservation and Germplasm Innovation in Mountainous Region (Ministry of Education), College of Life Sciences, Guizhou University, Guiyang 550025, China; 2Guizhou Province Key Laboratory of Agricultural and Animal Products Storage and Processing, School of Liquor and Food Engineering, Guizhou University, Guiyang 550025, China; 3Chili Pepper Research Institute, Guizhou Provincial Academy of Agricultural Sciences, Guiyang 550006, China; 4Department of Agricultural, Food and Nutritional Science, 4–10 Ag/For Building, University of Alberta, Edmonton, AB T6G 2P5, Canada

**Keywords:** fermented meat, microbial diversity, volatile organic compounds, correlation

## Abstract

Suan Rou (SR), a traditional fermented meat, is widely favored by consumers due to its unique flavor and characteristics. To study the relationship between the core differential micro-organisms and differential volatile organic compounds (VOCs) of SR from six regions of China, high-throughput sequencing (HTS) and gas-chromatography–ion mobility spectrometry (GC-IMS) technologies were used to analyze the correlation between micro-organisms and VOCs in SR from Xiangxi of Hunan, Rongshui of Guangxi, Zunyi of Guizhou, Jinping of Guizhou, Congjiang of Guizhou, and Libo of Guizhou. A total of 13 core micro-organisms were identified at the genus level. Moreover, 95 VOCs were identified in the SR samples by GC-IMS analysis, with alcohols, aldehydes, ketones, and esters comprising the major VOCs among all the samples. The results showed a strong correlation (|r| > 0.8, *p* < 0.05) between the core differential micro-organisms and differential VOCs, including four bacteria, five fungi, and 12 VOCs. *Pediococcus*, *Debaryomyces*, *Zygosaccharomyces*, and *Candida* significantly contributed to the unique VOCs of SR.

## 1. Introduction

Meat is a staple food in the human diet due to its high nutritional and biological values. As a traditional and important meat processing technology, fermentation provides meat with a longer shelf life [1,2,3], higher safety [4], better nutritional value, and a richer aroma and taste [5]. An accumulation of studies has reported that the unique flavor of fermented meat products is not only related to the enzymes in the raw meat but also closely related to the various micro-organisms that are involved in the natural fermentation process [6]. Micro-organisms affect the flavor of fermented meat products by participating in (1) the hydrolysis and self-oxidation of lipids in the raw meat [7,8], (2) the hydrolysis of proteins and carbohydrates [9,10,11], (3) the formation of amino acid and peptide derivatives with flavor activity [12], and (4) the formation of volatile organic compounds (VOCs) [13].

Suan Rou (SR) is a type of traditional fermented meat product with rich dietary and cultural connotations and unique regional and national characteristics. It is an important meat product for the Dong, Miao, and Buyi ethnic minorities in China, with a history of more than 2000 years. The product is prepared by mixing pork, salt, and other ingredients such as rice, pepper, chili, and garlic, and naturally fermenting them in an anaerobic environment for 1–2 months. The final product is widely favored by consumers due to its unique flavor, nutritional value, and non-greasy characteristics. However, the differences in environmental climate, production technology, raw materials, ingredients, and many other factors in different regions induce significant differentiation in the types and metabolic characteristics of microflora, thus forming their regional flavor characteristics. Therefore, researchers usually pay close attention to the microflora, protein, fat [8,14], flavor formation [15], and VOCs in fermented meat products [16], to ensure the stability of product quality and for the realization of industrial, modern, and large-scale production.

According to the region, SR can be classified into SR of Hunan, SR of Guangxi, and SR of Chongqing and Guizhou. Although there are relevant studies on the processing technology [17], flavor substances [18], and microbial colony structure [19,20], of Chinese traditional SR, these researches are mostly limited to the comparative analysis between the samples in the same or seldom regions [17,21]. Even if there is a comparison of products among provinces, only the impact of bacteria on VOCs has been analyzed, but the correlation analysis between core differential micro-organisms (bacteria and fungi) and differential VOCs is not deep enough to simultaneously compare samples from different provinces and regions of the same province. Therefore, 18 SR samples were collected from six representative regions: Xiangxi of Hunan, Rongshui of Guangxi, Zunyi of Guizhou, Jinping of Guizhou, Congjiang of Guizhou, and Libo of Guizhou, and their core differential micro-organisms and differential VOCs were analyzed by high-throughput sequencing (HTS) and gas-chromatography-ion mobility spectrometry (GC-IMS) techniques. The core differential micro-organisms and differential VOCs were determined by bidirectional orthogonal projections to latent structure-discriminant analysis (O2PLS-DA), and their interaction was evaluated by Spearman’s correlation coefficient analysis.

## 2. Materials and Methods

### 2.1. Sample Collection

A total of 18 SR samples were collected from local producers in six different representative producing regions of China. The raw and auxiliary materials of each sample group are listed in Table 1. The preparation methods of each group were similar: wash and slice the fresh pork, knead and decorate with 4–5% salt and 10% rice flour or millet according to local convention. These ratios were computed with the proportions of the raw meat. Next, the corresponding excipients were added, mixed through evenly, put into the fermentation container and sealed tightly to prevent the exchange of oxygen with the outside environment during the fermentation process, and naturally fermented for about 2 months according to the local natural environmental conditions. Finally, the completely fermented samples were collected and stored at −80 °C until use. Three samples that weighed 500 g each were collected.

### 2.2. DNA Extraction and Sequencing

#### 2.2.1. Total Genomic DNA Extraction and PCR Amplification

The total microbial genomic DNA was extracted from the SR samples using the FastDNA^®^ Spin Kit for Soil (MP Biomedicals, Irvine, CA, USA) according to the manufacturer’s instructions. The quality and concentration of DNA were determined by 1.0% agarose gel electrophoresis and NanoDrop^®^ ND-2000 spectrophotometry (Thermo Fisher Scientific, Inc., Waltham, MA, USA).

The primer pairs of the bacterial and fungal genes were 338F (5′-ACTCCTACGGGAGGCAGCAG-3′) and 806R (5′-GGACTACHVGGGTWTCTAAT-3′) [22], ITS1F (5′-CTTGGTCATTTAGAGGAAGTAA-3′), and ITS2R (5′-GCTGCGTTCTTCATCGATGC-3′) [23], respectively. The PCR reaction mixture is presented in Table 2. The PCR amplification cycling conditions were as follows: initial denaturation at 95 °C for 3 min, followed by 28 cycles of denaturing for bacteria and 40 cycles for fungi of denaturing at 95 °C for 30 s, annealing at 55 °C for 30 s, and extension at 72 °C for 45 s, and single extension at 72 °C for 10 min, and terminated at 4 °C. All samples were amplified in triplicate. The PCR product was extracted from a 2% agarose gel, purified using an AxyPrep DNA Gel Extraction Kit (Axygen Biosciences, Union City, CA, USA) according to the manufacturer’s instructions, and quantified using a Quantus™ Fluorometer (Promega, Madison, WI, USA).

#### 2.2.2. Illumina Sequencing and Bioinformatics Processing

The purified amplicons were pooled in equimolar amounts and paired-end sequenced on an Illumina MiSeq PE300 platform (Illumina, San Diego, CA, USA) according to the standard protocols reported by Majorbio Bio-Pharm Technology Co. Ltd. (Shanghai, China).

#### 2.2.3. Data Processing

The raw data were submitted to the NCBI Sequence Read Archive (SRA), with the accession number PRJNA860537. A bioinformatics analysis was performed using the Majorbio Cloud platform (https://cloud.majorbio.com (accessed on 1 May 2022)). The gene sequences from each sample were rarefied to 20,000, which yielded an average Good’s coverage of 99.99%. Based on the OTUs information, the rarefaction curves and alpha diversity indices, including the observed OTUs, Chao1 richness, Shannon index, and Good’s coverage were calculated using Mothur v1.30.1 (https://www.mothur.org/wiki/Download_mothur, accessed on 1 May 2022). The similarity among the microbial communities in different samples was determined by a principal coordinate analysis (PCoA) based on Bray-Curtis dissimilarity using the R’s Vegan v2.5-3 package. The percentage of variation explained by the treatment was assessed by a PERMANOVA test, and its statistical significance was determined using the Vegan v2.5-3 package. The significantly abundant taxa (phylum to genera) of bacteria among the different groups (LDA score > 4, *p* < 0.05) were identified by the linear discriminant analysis (LDA) effect size (LEfSe) (http://huttenhower.sph.harvard.edu/LEfSe (accessed on 1 May 2022)).

### 2.3. Analysis of Volatile Compounds

The VOCs in each sample group were detected using the GC-IMS flavor analyzer (FlavourSpec ^®^, G.A.S. Dortmund, Germany). The procedure was as follows: The sample (2 g) was ground at a low temperature and placed in a 20 mL headspace vial, incubated at 60 °C for 20 min, and then the 500 μL headspace sample was automatically injected into the injector at 85 °C through a heated syringe (non-shunt mode). The sample was transferred into an MXT-5 capillary column (15 m × 0.53 mm, film thickness 1 μm) (Restek, Bellefonte, PA, USA) through high-purity nitrogen (99.99%) and introduced into an ionization chamber after elution at 60 °C (isothermal mode). The sample was later scanned in the drift tube, and each spectrum was scanned 12 times. The VOCs were identified by comparing their retention index and drift time with the standards in the GC-IMS library. The relative quantification of VOCs was based on the peak signal intensity. The fingerprint of VOCs was constructed using GalleryPlot (FlavourSpec ^®^, G.A.S. Dortmund, Germany) supported by a GC-IMS instrument.

### 2.4. Statistical Analysis

The statistical and Spearman’s correlation analyses were performed using the Statistical Package for the Social Sciences (SPSS 19.0, IBM, Inc., Armonk, NY, USA), and the significances among the groups were evaluated using the multiple comparative analysis of variance (ANOVA). The microbial data were analyzed on the online platform of Majorbio Cloud Platform (www.majorbio.com (accessed on 1 May 2022)). The major flavor compounds were screened and analyzed by the principal component analysis (PCA), O2PLS-DA, and permutations plot using SIMCA^®^ (version 14.1, Sartorius Stedim Data Analytics AB, Umeå, Sweden). The results of a Spearman’s correlation analysis indicated that the relationship between the micro-organisms and flavor substances was established by Gephi (0.9.25) (https://gephi.org/ (accessed on 1 May 2022)), and the network map was optimized using the Cytoscape (3.9.1) software (http://www.cytoscape.org/ (accessed on 1 May 2022)). All the experiments were performed in triplicate, and the data are expressed as the means ± standard deviations.

## 3. Results and Discussion

### 3.1. Microbial Community Characteristics of SR from Different Regions

#### 3.1.1. Abundance and Diversity of Bacterial and Fungi

Two high-quality sequences, i.e., 1,236,441 16s rRNA and 1,232,142 ITS1 were obtained from 18 SR samples by HTS. The sequences of all the samples were clustered according to the 97% similarity level, and 279 OTUs of bacteria and 846 OTUs of fungi were obtained. The dilution curve and Shannon index curve were close to the saturation platform, indicating that the sequencing data could cover almost all the micro-organisms in the sample, which is sufficient for subsequent analysis (Figure 1A,B). The alpha diversity of the microbial communities in the SR samples was represented by the Shannon, Simpson, Chao1, and ACE indices. A higher Shannon index indicates a higher alpha diversity of the microbial community in the SR samples, while the Simpson index had the opposite response The Chao1 and ACE indices represent the richness of microflora, and a higher value indicates that the microflora is richer. The α-diversity index is summarized in Table 3. There was no significant difference in the Shannon, Simpson, ACE, and Chao1 indices among the different groups of bacteria, indicating that all the samples showed similar trends in bacterial richness and diversity. There were some significant differences in the α-diversity index among the different groups of fungi, with HNXX showing the highest fungal diversity and GZZY showing the highest fungal richness. The β-diversity of the bacteria and fungi in each sample group was analyzed by NMDS and PCoA analyses. The variability and similarity of the microbial population structure in different SR samples are illustrated in Figure 1C,D.

##### Microbial Composition Analysis

The sequencing data of bacteria and fungi were analyzed at the phylum and genus levels to further study the community structure of each sample (Figure 2). A total of 12 bacterial phyla and 7 fungal phyla were identified at the phylum level in the 18 SR samples. Only those with microbial abundance greater than 1% are depicted in Figure 2, and the rest were merged with the others. The bacterial communities of the samples mainly included *Firmicutes* bacteria, and the proportion in each group was more than 95%, followed by *Proteobacteria* and other bacteria with a microbial abundance of less than 1% (Figure 2A). The results were consistent with those reported by Lv [12], and Wang [24]. *Firmicutes* accounted for the highest proportion in GXRS (99%), followed by GXJP (98%), GZCJ (98%), GZZY (97%), GZLB (96%), and HNXX (95%). Except for GZCJ, the fungal communities of all the samples were mainly *Ascomycota*, followed by *Basidiomycota*, unclassified fungi, and other fungi with microbial abundances less than 1%, and the proportion of *Ascomycota* and *Basidiomycota* was more than 93% (Figure 2B). However, the dominant fungi in GZCJ were *Basidiomycota* (59%) and *Ascomycota* (41%) (Figure 2B).

A total of 130 bacterial genera were identified in 18 SR samples, with *Lactobacillus*, *Weissella*, *Lactococcus*, *Leuconostoc*, *Pediococcus*, *Staphylococcus*, *Enterobacter*, *Teragenococcus*, and *Macrococcus* comprising the core genera (opposite abundance top 10) (Figure 2A). *Lactobacillus* had the highest abundance in GXRS (98.18%), followed by GZZY (81.13%), GZCJ (73.39%), GZLB (72.86%), HNXX (47.56%), and GZJP (28.80%). *Lactobacillus* can release various enzymes during growth and metabolism and convert the substrates in raw meat into the aroma and flavor substances of fermented meat products [25]. The relative abundance of *Weissella* detected in GZJP was 61.21%, which was much higher than that in other samples. The relative abundance of *Lactococcus* in HNXX (40.37%) was the highest, while the relative abundance of *Leuconostoc* in GZZY (7.56%) was higher than that in other groups. The relative abundance of *Staphylococcus* in GZLB and GZCJ was relatively high, accounting for 2.02% and 2.21%, respectively, while the relative abundance of *Pediococcus* in HNXX, GZLB, and GZZY was 1.71%, 2.04%, and 2.29%, respectively. Notably, the relative abundance of *Staphylococcus* and *Pediococcus* in the GZLB samples was higher than that in the other groups, and these two bacteria could mainly improve the flavor mainly through the metabolism of proteins and lipids [6,26]. Other core bacteria also existed at different proportions in all the SR samples.

A total of 291 fungal genera were identified in all the samples, and the top ten most abundant genera were *Wallemia*, *Aspergillus*, *Kodamaea*, *Candida*, *Gibberella*, *Debaryomyces*, *Zygosaccharomyces*, *Ogataea*, *Kazachstania*, and *Cystofilobasidium* (Figure 2B). *Debaryomyces* and *Candida* were the common fungi in fermented meat products, which could stabilize the color of meat products through their deoxygenation ability and promote the flavor formation of fermented meat products by decomposing the lipids and proteins through the enzymes [27]. *Debaryomyces* had the highest relative abundance in GZLB (14.47%), followed by GXRS (7.66%), which has been widely used as an auxiliary starter. It is not only a beneficial fungus in Panxian ham [28], but also an important fungus in the fermentation and ripening of sausage [29]. *Candida* had the highest relative abundance in GZZY (19.53%), and it plays a vital role in the formation of meat flavor and can promote the decomposition of protein in fermented meat products [30]. Mi [31], reported that *Kazachstania* was one of the core fungi in sour meat. In some fermented foods, *Kazachstania* could produce VOCs, which play a vital role in the formation of fermented food flavor. It was reported that *Aspergillus* has excellent antibacterial activity and antioxidant properties in meat products [32], with the highest content in HNXX (34.89%).

##### Microbial Difference Analysis

The micro-organisms with significant abundance differences among the SR samples were detected using the non-parametric factorial Kruskal-Wallis (KW) sum-rank test. The LEfSe analysis (Figure 3) results showed that the differences in the genus level of each group were mainly composed of 29 microbial species (7 species of bacteria and 22 species of fungi), 5 core bacterial species (*Enterobacter*, *Lactococcus*, *Weissella*, *Lactobacillus*, and *Pediococcus*), and 8 core fungal species (*Wallemia*, *Aspergillus*, *Kodamaea*, *Candida*, *Gibberella*, *Debaryomyces*, *Zygosaccharomyces*, *Cystofilobasidium*). *Weissella*, *Pediococcus*, *Lactococcus*, *Lactobacillus*, *Debaryomyces*, *Candida*, *Kodamaea*, and *Gibberella* were found that could enhance the aroma characteristics of fermented food. Therefore, their abundance differences in different samples might directly or indirectly affect the composition of VOCs.

The analysis also revealed that there were some differences in the microbial composition of SR samples from different provinces and different regions of the same province, which might have been influenced by the raw materials, ingredients, production environment, fermentation temperature, and relative humidity.

### 3.2. Flavor Compounds Analysis

GC-IMS is an effective method for the separation and sensitive detection of VOCs [33]. The VOCs in 18 SR samples were detected by the GC-IMS technique. A total of 104 VOCs were detected and 95 were identified, including 84 monomer compounds and 11 monomer polymer compounds. These compounds were composed of 21 aldehydes, 19 esters, 16 alcohols, 11 ketones, 8 alkenes, 2 acids, and 7 other compounds (Table 4). As shown in Table 4, the composition of VOCs was generally similar among the samples, but the content was different. Overall, the alcohols, ketones, and aldehydes in Guizhou SR were higher than those in Hunan and Guangxi, while esters were on the contrary. Acids in GZJP were the highest, followed by HNXX, GZCJ, GXRS, GZZY, and GZLB. Alkenes in GXRS were the highest among the samples. These differences may be caused by the differences in environmental climate, raw materials, and many other factors in different regions. According to the PCA (Figure 4A,B) and GalleryPlot (Figure 4C) results, there were differences in the VOCs among the samples of each group, indicating that the composition and proportion of VOCs in the SR samples from different provinces and different regions of the same province were different.

The differences in the abundance of VOCs in these samples were then detected by O2PLS-DA. The Permutations Plot helps to assess the risk that the model is spurious. The results of the permutations plot were shown that all R2-values and Q2-values to the left are lower than the original points to the right, and the regression line of the Q2-points intersects below zero, which means the original model was valid (Appendix A). The influence of each difference in metabolite accumulation on each sample classification and its explanatory power was investigated according to the projection variable importance (VIP) score. VIP ≥ 1 was the screening criteria for common differential metabolites [34]. A total of 42 VOCs were found to have significant differences among the samples (VIP > 1, *p* < 0.05) (Figure 4D), including 8 alcohols, 7 ketones, 10 esters, 10 aldehydes, 2 acids, 2 olefins, and 3 other compounds.

Alcohols are the essential flavor components and are closely related to lipid oxidation, amino acid metabolism, methyl ketone reduction, and microbial reproduction [35]. In this study, eight types of alcohols were detected with significant differences among the samples, including 1-penten-3-ol, (Z)-2-pentenol, 1-pentanol, 1-hexanol, 1-octen-3-ol, 1-octanol, linalool, and cis-*p*-menth-2-en-1-ol. The content of 1-pentanol was relatively high in the GZZY, GZCJ, and GZLB samples, while the relative content of 1-hexanol in the GZZY samples was significantly higher than that in the other groups. Linalool was abundant in the GZJP and GZCJ samples and significantly differed from the other groups. 1-octen-3-ol is a common unsaturated alcohol in fermented meat products with a low odor threshold that is oxidized by arachidonic acid and has a mushroom and flower scent [36]. In this study, 1-octen-3-ol was found in all samples, but its content was high in GZZY, which was consistent with the results of Wang [24]. Previous studies have reported that 1-hexanol and 1-octen-3-ol are one of the primary flavor components of Dong sour meat [19,24].

Ketones, the primary source of animal and plant fat flavor, can be produced by automatic lipid oxidation and microbial metabolism. In this study, the 7 ketones, namely 2-propanone, 2-butanone, 2-pentanone, 3-hydroxy-2-butanone, 2-hexanone, dihydro-2-methyl-3(2H) furanone, and coumarin showed significant differences among all the samples. Among them, the contents of 2-propanone and 2-butanone were relatively high. However, the contents of dihydro-2-methyl-3(2H) furanone and coumarin in the HNXX samples were significantly higher than those in the other groups, showing a more abundant ketone flavor.

Esters were the primary VOCs in all SR samples with a special fruit flavor. After meat fermentation, the short-chain acids were esterified with alcohol to form the esters. In this study, 10 different esters were detected, including 1 polymer, such as methyl acetate, ethyl acetate, propyl acetate, ethyl 2-methyl propanoate, ethyl 3-methyl butanoate, propyl butanoate, ethyl hexanoate, ethyl 2-hydroxy-4-methyl pentanoate, and geranyl acetate. Among them, ethyl acetate, propyl acetate, ethyl 2-methyl propanoate, and ethyl 3-methyl butanoate were abundant. Ethyl hexanoate was detected in all the samples and showed significant differences among the samples. It has been reported that ethyl hexanoate is the main flavor substance of Dong sour meat [31]. These ethyl esters contribute to the fruit and creamy flavor of sour meat, thereby promoting the formation of a sour meat flavor quality.

Aldehydes, another essential flavor substance, are mainly derived from the oxidation of unsaturated fatty acids. The odor threshold of these substances is low and most of them have a fruity aroma, contributing to the overall flavor of processed meat products [31]. In this study, the 10 aldehydes, namely butanal, 2-methylbutanal, pentanal, (E)-2-pentenal, hexanal (dimer), benzaldehyde, 2,4-heptadienal, octanal, benzeneacetaldehyde, and nonanal showed significant differences among the SR samples. Hexanal is obtained from the oxidation of n-6 fatty acids (oleic acid and arachidonic acid) with a strong raw fat flavor, which is a unique flavor substance in fresh meat. It could be an indicator of the oxidation level in the fermented meat and imparts a green grass odor [37]. However, excessive hexanal can lead to rotten odors, while nonanal and other linear aldehydes contribute to a sour meat flavor.

The volatile acids in fermented meat are mainly produced by the hydrolysis of phospholipids and triglycerides and lipid oxidation. The low odor threshold of short-chain acid (C < 6) contributes to the formation of aroma and flavor characteristics in fermented meat, thus affecting the flavor formation of fermented meat products [38]. In contrast to other previous reports [18,24], only two volatile acids, i.e., propanoicacid and 3-methylbutanoicacid, were detected in this study. Propanoic acid was only found in the GZJP, GZZY, and GZCJ samples. This result might be due to the esterification of acids with alcohols to form esters, resulting in low contents of acids that were below the detection line and undetectable. The decrease in the acidity of the sample depends on the realization of non-volatile acids, which can be inferred from the type and content of esters.

Among the alkenes, alpha-pinene and gamma-terpinene with citrus and lemon aroma were significantly different in the 18 samples, providing abundant flavor to SR. Other flavor compounds formed by the maillard reaction during the fermentation were pyrazine, furan, ether, and sulfur compounds. Among these volatile compounds, 3 kinds of volatile compounds showed significant differences, including dimethyl disulfide, 2,3-diethyl-5-methyl pyrazine, and methyl chavicol. However, dimethyl disulfide was not detected in GZZY and GZLB.

### 3.3. Co-Occurrence and Exclusion Analyses Revealed the Relationships between Different Microbes

Microbial interactions are essential factors that affect the microbial structure. The interaction between the micro-organisms was investigated by Spearman’s correlation coefficients and *p*-values to construct a network diagram of the core bacteria and fungi (Figure 5A). According to the correlation analysis results, the interaction between the bacteria was more abundant than that in the fungi, and there was a positive correlation between *Pediococcus* and *Lactococcus*; *Klebsiella*, *Enterobacter*, and *Macrococcus* (|r| > 0.6, *p* < 0.05). *Teragenococcus*, *Weissella*, and *Staphylococcus* were in a mutually reinforcing relationship with each other, and *Teragenococcus* was negatively correlated with *Enterobacter*, *Klebsiella*, *Lactobacillus*, and *Lactococcus* (|r| > 0.6, *p* < 0.05). There was a positive correlation between *Enterobacter*, *Klebsiella*, and *Lactococcus*; and *Macrococcus* and *Leuconostoc*, while there was a negative correlation between *Lactobacillus* and *Weissella* (|r| > 0.6, *p* < 0.05). As for the fungi, *Candida* showed a positive relationship with *Wallemia* and *Zygosaccharomyces*, but a negative relationship with *Debaryomyces* and *Aspergillus* (|r| > 0.6, *p* < 0.05). There was a positive correlation between *Aspergillus* and *Zygosaccharomyces*, but a negative correlation between *Gibberella* and *Alternaria* (|r| > 0.6, *p* < 0.05).

The network diagram indicates that the bacteria and fungi share a close relationship with each other. *Candida* and *Debaryomyces* had the most abundant relationship with bacteria, which positively correlated with *Enterobacter*, *Klebsiella*, *Macrococcus*, and *Lactococcus*, but negatively correlated with *Tetragenococcus* and *Weissella* (|r| > 0.6, *p* < 0.05). *Candida* and *Aspergillus*, *Debaryomyces* and *Cystofilobasidium*, *Kodamaea*, and *Gibberella* positively correlated with *Pediococcus*, *Lactobacillus*, *Staphylococcus*, and *Leuconostoc* (|r| > 0.6, *p* < 0.05), respectively. *Wallemia* negatively correlated with *Enterobacter* and *Wallemia* and *Zygosaccharomyces* negatively correlated with *Macrococcus* and *Pediococcus* (|r| > 0.6, *p* < 0.05). *Cystofilobasidium*, *Gibberella*, and *Ogataea* negatively correlated with *Weissella*, *Lactococcus*, and *Lactobacillus* (|r| > 0.6, *p* < 0.05), respectively.

### 3.4. Correlations between Micro-Organisms and Volatile Organic Compounds

The interaction between VOCs and major micro-organisms (relative abundance top 10) in the SR samples from different regions was studied using Spearman’s correlation coefficient. The results showed that 8 bacterial species and 9 fungal species shared a significant correlation with 82 species and 67 species of VOCs, respectively (|r| > 0.6, *p* < 0.05). As shown in Figure 5B that the bacteria associated with less than 10 species of VOCs were *Leuconostoc* (9 species), and the fungi were *Alternaria* (1 species), *Gibberella* (3 species), and *Kodamaea* (8 species). The bacteria associated with more than 10 species of VOCs were *Staphylococcus* (19 species), *Tetragenococcus* (18 species), and *Lactococcus* (18 species), and fungi were *Aspergillus* (14 species), *Ogataea* (14 species), *Wallemia* (18 species), and *Zygosaccharomyces* (19 species). The bacteria associated with more than 20 species of VOCs were *Pediococcus* (27 species), *Weissella* (25 species), *Lactobacillus* (25 species), and *Enterobacter* (23 species), and fungi were *Candida* (21 species), and *Debaryomyces* (22 species). Although there were more fungal species related to VOCs than bacteria, fine bacteria could affect more VOCs than fungi.

Additionally, the correlation between the core differential micro-organisms (five bacteria and eight fungi) and 42 differential VOCs (VIP ≥ 1) was analyzed. Finally, five bacterial genera (*Lactococcus*, *Pediococcus*, *Weissella*, *Lactobacillus*, and *Enterobacter*) and seven fungal genera (*Candida*, *Debaryomyces*, *Gibberella*, *Wallemia*, *Aspergillus*, *Zygosaccharomyces* and *Gibberella*) were correlated with 35 species and 35 species of VOCs (|r| > 0.6, *p* < 0.05), respectively (Figure 5C). As for bacteria, *Enterobacter*, *Weissella*, *Lactococcus*, *Lactobacillus*, and *Pediococcus* were related to 14, 12, 11, 10, and 6 types of VOCs, respectively. *Lactobacillus*, *Lactococcus*, and *Weissella* are beneficial to human health and are often used as starters in fermented products. *Weissella* was positively correlated with 9 types of VOCs, followed by *Lactococcus* (three kinds), while *Lactobacillus* was negatively correlated with 10 types of VOCs (|r| > 0.6, *p* < 0.05). They play a key role in flavor formation and can increase the content of some metabolites, such as acids and alcohols [39]. These metabolites were positively correlated with methyl acetate, linalool, and cis-*p*-menth-2en-1-ol (|r| > 0.6, *p* < 0.05). These bacteria have been widely used in the production of fermented food due to their ability to increase the content of organic acids, short-chain fatty acids, and esters. *Pediococcus* plays an important role in the formation of the final flavor quality of fermented meat products [40]. In this study, *Pediococcus* had a significantly positive correlation with three volatile compounds, including hexanol, methyl acetate, and benzeneacetaldehyde, but a significantly negative correlation with ethyl 3-methyl butanoate, hexanal, and dimethyl disulfide (|r| > 0.6, *p* < 0.05). This correlation might be related to their ability to produce high contents of protease, which promote the hydrolysis of proteins to produce free amino acids, thereby contributing to the formation of VOCs [41]. Although *Enterobacter* was correlated with 14 types of VOCs, most were negatively correlated with 12 species but positively correlated with 1-penten-3-ol and octanal (r > 0.6, *p* < 0.05). Notably, although *Staphylococcus* and *Tetragenococcus* affected 42 types of differential VOCs, there was no difference in the bacterial composition of each sample. Therefore, it was inferred that these bacteria might have contributed to the flavor composition of the sample, but they were not the primary micro-organisms affecting the flavor differences.

The fungi *Debaryomyces* and *Wallemia* correlated with 11 and 8 types of VOCs, respectively. *Aspergillus*, *Kodamaea*, and *Zygosaccharomyces* were associated with seven types of VOCs. *Candida* and *Gibberella* were associated with six and two types of VOCs (r > 0.6, *p* < 0.05), respectively. *Aspergillus*, *Candida*, and *Debaryomyces* are the essential factors in fermented meat products [42]. The primary VOCs of SR come from the decomposition of proteins and the transformation of amino acids by yeast. Most yeasts were positively correlated with alcohols and esters, while *Aspergillus* and *Candida* promoted the production of higher alcohols, acetates, and fatty acid esters. *Debaryomyces* was positively correlated with esters and aldehydes but negatively correlated with higher alcohols and ketones (r > 0.6, *p* < 0.05). This might be because *Debaryomyces* promotes esterification and oxidation, thus consuming alcohols to form esters. *Wallemia* and *Zygosaccharomyces* were positively correlated with six and three types of VOCs and negatively correlated with two and four types of VOCs, respectively. They also contributed to the composition of meat flavor substances, and *Zygosaccharomyces* could release important flavor compounds [43], such as fusel alcohols and the derivatives of 4-hydroxyfuranone. These compounds have soy sauce and smoked flavors, and contribute to the composition of flavor substances in the sample. Although *Ogataea* had some effect on different flavor substances, there was no significant difference in each sample group. Therefore, it might have contributed somewhat to the flavor of samples, but cannot affect the differences in VOCs among the samples.

After further stringent requirements on the correlation coefficient, a strong correlation was found between the core differential micro-organisms and differential VOCs (Figure 5D), including four bacteria, five fungi, and 12 VOCs (|r| > 0.8, *p* < 0.05). Methyl acetate and nonanal, which contributed to the flavor of SR [44], were significantly influenced by the bacteria and fungi. The bacteria had a strong correlation with one alcohol, one ketone, one ester, two aldehydes, and one alkene. *Pediococcus* and *Lactobacillus* could affect more VOCs than other bacteria, and *Pediococcus* had a positive correlation with 1-hexanol, which is the primary substance of SR flavor [19,24]. However, other bacteria had a negative correlation with the VOCs. The results showed that *Pediococcus* had a positive contribution to the formation of the characteristic flavors of SR in different regions. The fungi were strongly correlated with one ketone, three esters, three aldehydes, and one disulfide. It was also observed that the fungi mainly affected the esters and aldehydes. *Debaryomyces* could affect more VOCs than the other fungi, followed by *Zygosaccharomyces*. It is noteworthy that *Candida* has a negative correlation with Dimethyl disulfide with an unpleasant odor and contributes to the formation of the special flavor of SR.

## 4. Conclusions

This study reports the differences in microbial community composition and VOCs of SR from six regions of China. The analysis of micro-organisms showed that the core differential micro-organisms were primarily composed of five bacterial species and eight fungal species. The bacteria had a significant effect on ketones and esters (7 species and 10 species, respectively), while the fungi had a significant effect on alcohols and aldehydes (5 species and 7 species, respectively). There was a strong correlation between nine core differential micro-organisms and 12 differential VOCs. *Pediococcus*, *Debaryomyces*, *Zygosaccharomyces*, and *Candida* had positive effects on the formation of the special VOCs of SR. The relationship between the core differential micro-organisms and differential VOCs provides a strong basis for the further study of VOCs from the microbial ecology of traditional fermented meat products. Proteomics and other multi-group methods combined with the threshold of VOCs are the potential methods for exploring the relationship between the VOCs. In addition, determining the key aroma compounds of SR and relative microbial metabolic pathways could help broaden the industrial production of traditional SR.

## Figures and Tables

**Figure 1 foods-11-02708-f001:**
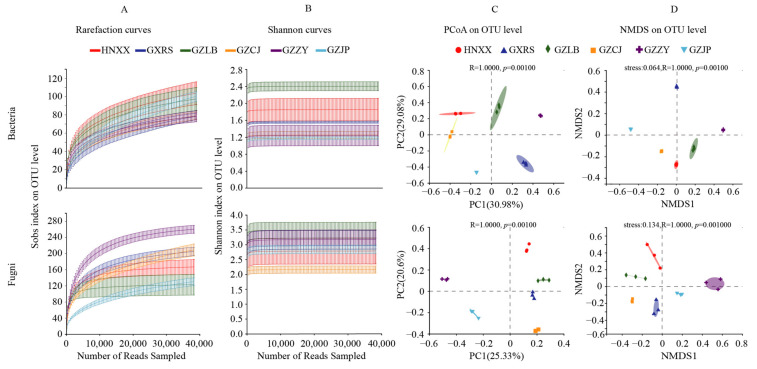
Variations in the microbial diversity and community structure of SR from different regions. (**A**) Rarefaction curves of bacteria and fungi for each sample. (**B**) Shannon index curves of bacteria and fungi for each sample. (**C**,**D**) PcoA score plots and NMDS of bacteria and fungi.

**Figure 2 foods-11-02708-f002:**
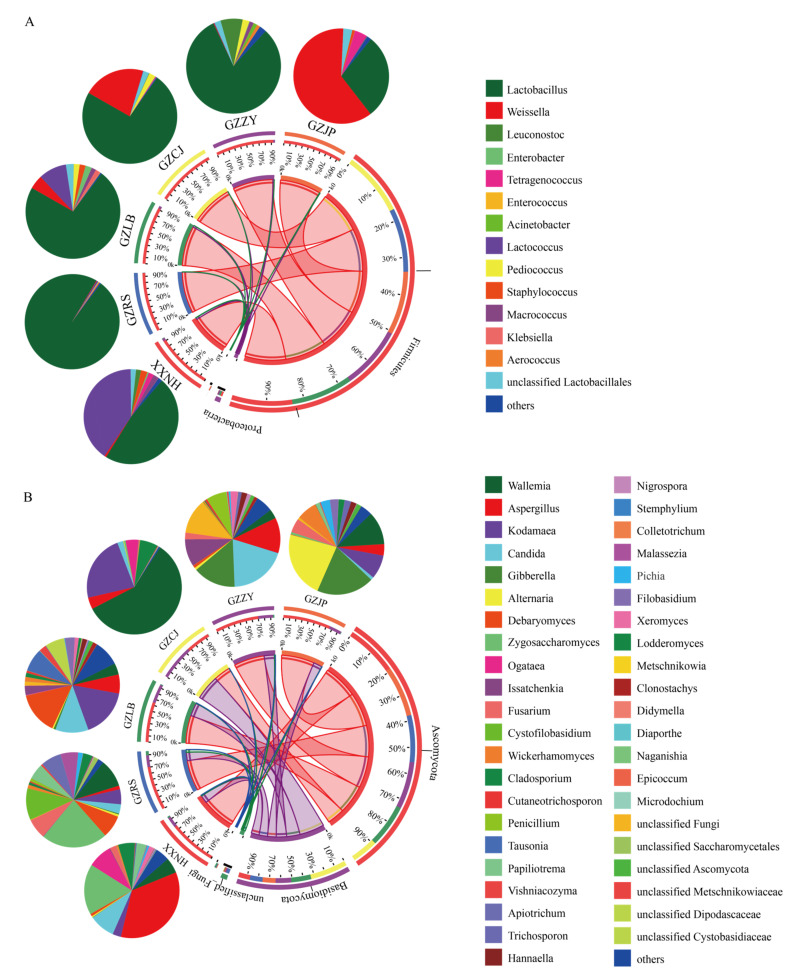
Relative abundance of bacteria at the phylum and genus level (**A**) and fungi at the phylum and genus level (**B**) of SR from different regions.

**Figure 3 foods-11-02708-f003:**
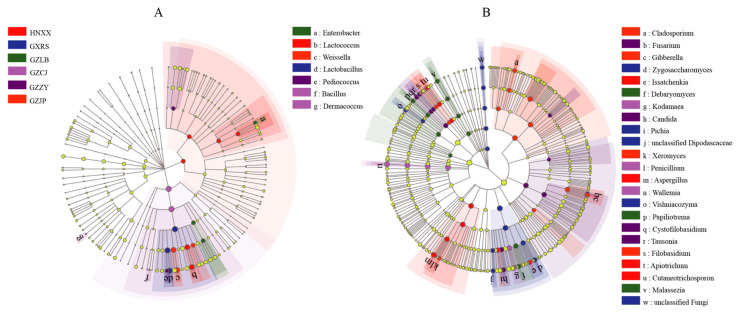
LEfSe analysis diagram of each dominant taxa of bacteria (**A**) and fungi (**B**) during the fermentation of SR. Nodes with different colors indicate microbial groups that are significantly enriched in the corresponding groups and have a significant impact on the differences between groups; light yellow nodes indicate that there are no significant differences in different groups (*p* > 0.05). The legend on the right shows micro-organisms that have changed significantly at the genus level (*p* < 0.05).

**Figure 4 foods-11-02708-f004:**
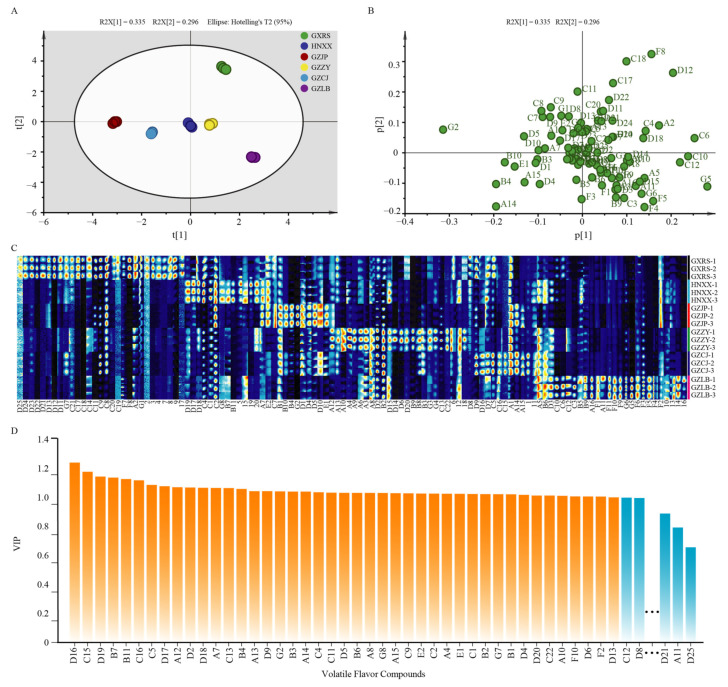
Volatile organic compounds analysis of SR from different regions. (**A**) Principal component analysis (PCA) and (**B**) load diagram results for flavor metabolite contents in SZR samples. (**C**) Gallery Plot of SR from different regions. Each line in the figure represents all signal peaks selected from a sample. The brighter the color is, the stronger the signal is, and the darker the color is, the weaker the signal is. (**D**) The variation of VIP (pred) values of VOCs in SR samples from different regions. Orange indicates VOCs with VIP > 1 and blue indicates VOCs with VIP < 1. All the VOCs with VIP > 1 and part of the VOCs with VIP < 1 are shown in the figure.

**Figure 5 foods-11-02708-f005:**
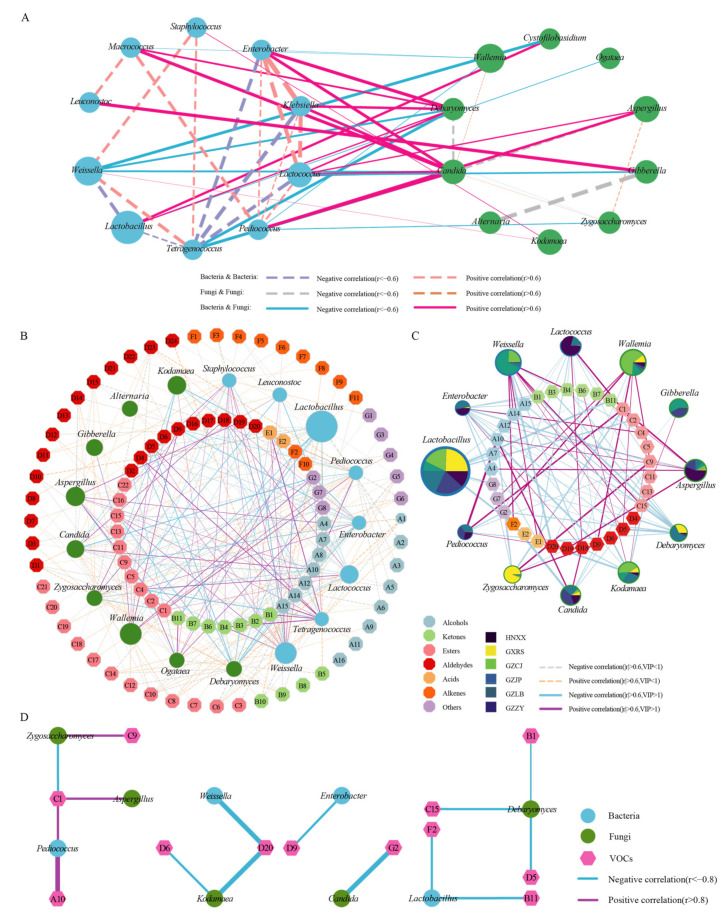
Correlation analysis. Statistical significance (*p* < 0.05), Spearman correlation coefficient (|r| > 0.6, |r| > 0.8) represents correlation. (**A**) Association network diagram of bacteria and fungi. The orange and green circles refer to bacteria and fungi, respectively, and the purple and blue lines refer to positive correlations (r > 0.6 and *p* < 0.05) and negative correlations (r < −0.6 and *p* < 0.05), respectively. (**B**) Correlation of major differential micro-organisms with all VOCs. The blue and green dots represent bacteria and fungi respectively, and the dot size is proportional to their relative abundance. Different VOC classifications are shown in different colors, with solid and dotted lines indicating positive correlations (r > 0.6 and *p* < 0.05) and negative correlations (r < −0.6 and *p* < 0.05), respectively. The thickness of each connection (edge) between the two dots is directly proportional to the value of the Spearman correlation coefficient. (**C**) Correlation between major differential micro-organisms and differential VOCs (VIP > 1). The blue and green circles represent bacteria and fungi respectively. The node size is in a positive proportion to its relative abundance. The pie chart represents the relative abundance of the micro-organism in different samples. Different VOC classifications are shown in different colors, with solid and dotted lines indicating positive correlations (r > 0.6 and *p* < 0.05) and negative correlations (r < −0.6 and *p* < 0.05), respectively. The thickness of each connection (edge) between the two nodes is directly proportional to the value of the Spearman correlation coefficient. (**D**) Correlation between core differential micro-organisms and differential VOCs. The blue and green circles represent bacteria and fungi respectively. The pink hexagon represents different VOCs. The solid lines indicate positive and negative correlations (|r| > 0.8, *p* < 0.05).

**Table 1 foods-11-02708-t001:** Information about the Suan Rou samples tested in this study.

Sample	Producers	Production Location	Raw Material	Fermented Conditions
HNXX	Local farmer 1	Xiangxi Tujia and Miao Autonomous Prefecture, Hunan Province	Pork belly, Salt, Rice, Pepper	A sealed fermented LNEC
GXRS	Local farmer 2	Rongshui Miao Autonomous County, Liuzhou City, Guangxi Zhuang Autonomous Region	Pork belly, Salt, Chili	A sealed fermented LNEC
GZZY	Local farmer 3	Zunyi City, Guizhou Province	Pork belly, Salt, Rice	A sealed fermented LNEC
GZLB	Local farmer 4	Libo County, Qiannan Buyi and Miao Autonomous Prefecture, Guizhou Province	Pork belly, Salt, Millet, Pepper	A sealed fermented LNEC
GZCJ	Local farmer 5	Congjiang County, Southeast Guizhou Miao and Dong Autonomous Prefecture, Guizhou Province	Pork belly, Salt, Millet, Chili, Pepper	A sealed fermented LNEC
GZJP	Local farmer 6	Jinping County, Southeast Guizhou Miao and Dong Autonomous Prefecture, Guizhou Province	Pork belly, Salt, Chili, Pepper	A sealed fermented LNEC

Note: LNEC means local natural environment conditions.

**Table 2 foods-11-02708-t002:** The PCR reaction mixture of bacteria and fungi.

	Bacteria	Fungi
Template DNA	10.0 ng	10.0 ng
Primer F (5.0 μM)	1.0 μL	0.80 μL
Primer R (5.0 μM)	1.0 μL	0.80 μL
dNTPs (2.5 mM)	2.0 μL	2.0 μL
Fast Pfu polymerase	0.50 μL	0.20 μL
5× Fast Pfu buffer	4.0 μL	2.0 μL
ddH_2_O	To a final volume of 20.0 µL	

**Table 3 foods-11-02708-t003:** Richness and diversity of bacteria and fungi of SR from different regions.

Simple ID	Shannon		Simpson		ACE		Chao1	
	Bacteria	Fungi	Bacteria	Fungi	Bacteria	Fungi	Bacteria	Fungi
HNXX	2.08 ± 0.45 ^a^	3.50 ± 0.21 ^a^	0.23 ± 0.08 ^a^	0.06 ± 0.02 ^e^	135.91 ± 21.93 ^a^	136.11 ± 18.66 ^c^	136.15 ± 6.39 ^a^	137.83 ± 25.86 ^c^
GXRS	1.78 ± 0.39 ^a^	1.30 ± 0.36 ^c^	0.29 ± 0.09 ^a^	0.37 ± 0.09 ^c^	187.52 ± 43.89 ^a^	138.74 ± 20.28 ^c^	153.54 ± 19.70 ^a^	129.51 ± 25.23 ^c^
GZCJ	1.71 ± 0.48 ^a^	0.28 ± 0.03 ^d^	0.36 ± 0.16 ^a^	0.89 ± 0.00 ^a^	135.44 ± 32.43 ^a^	68.06 ± 12.70 ^d^	125.69 ± 27.57 ^a^	45.67 ± 17.33 ^d^
GZLB	1.94 ± 0.40 ^a^	3.22 ± 0.21 ^a,b^	0.25 ± 0.08 ^a^	0.09 ± 0.03 ^d,e^	148.85 ± 15.66 ^a^	217.38 ± 12.42 ^b^	145.66 ± 1.19 ^a^	220.91 ± 16.50 ^b^
GZZY	2.12 ± 0.27 ^a^	3.18 ± 0.27 ^a,b^	0.25 ± 0.11 ^a^	0.10 ± 0.03 ^d,e^	147.49 ± 10.41 ^a^	276.27 ± 13.84 ^a^	150.16 ± 18.67 ^a^	277.17 ± 12.55 ^a^
GZJP	1.57 ± 0.07 ^a^	0.82 ± 0.26 ^e^	0.41 ± 0.07 ^a^	0.69 ± 0.11 ^b^	134.76 ± 22.91 ^a^	89.12 ± 22.06 ^d^	130.16 ± 14.58 ^a,b^	78.33 ± 15.93 ^d^

Values are presented as the mean ± standard deviation of quintuplicate (*n* = 3). Different letters in the same column represent significant differences (*p* < 0.05).

**Table 4 foods-11-02708-t004:** GC-IMS detected volatiles of SR from different regions.

NO.	Compound Name	Formula	CAS	Relative Content (%)					
				GZRS	HNXX	GZJP	GZZY	GZCJ	GZLB
Alcohols (16)				11.81 ± 0.07 ^f^	17.97 ± 0.16 ^d^	19.9 ± 0.34 ^c^	29.38 ± 0.24 ^a^	25.1 ± 0.08 ^b^	16.52 ± 0.43 ^e^
A1	Ethanol	C64175	C_2_H_6_O	4.96 ± 0.1987 ^e^	5.70 ± 0.1975 ^d^	10.29 ± 0.1351 ^a^	8.24 ± 0.1165 ^c^	9.21 ± 0.0862 ^b^	2.35 ± 0.1598 ^f^
A2	Propanol	C71238	C_3_H_8_O	2.81 ± 0.0364 ^a^	0.90 ± 0.0358 ^e^	0.28 ± 0.0220 ^f^	2.59 ± 0.0773 ^b^	1.06 ± 0.0414 ^d^	1.49 ± 0.0473 ^c^
A3	Isobutanol	C78831	C_4_H_10_O	0.10 ± 0.0013 ^e^	0.21 ± 0.0169 ^d^	0.17 ± 0.0051 ^d^	0.94 ± 0.0399 ^a^	0.56 ± 0.0225 ^c^	0.66 ± 0.0528 ^b^
A4	1-penten-3-ol	C616251	C_5_H_10_O	0.72 ± 0.0137 ^f^	2.87 ± 0.1810 ^b^	1.65 ± 0.0484 ^d^	4.03 ± 0.0317 ^a^	1.91 ± 0.0611 ^c^	0.95 ± 0.0507 ^e^
A5	Isopentanol	C123513	C_5_H_12_O	0.84 ± 0.0378 ^c^	2.91 ± 0.0411 ^b^	0.34 ± 0.0361 ^d^	2.94 ± 0.1447 ^b^	2.96 ± 0.0426 ^b^	3.53 ± 0.0784 ^a^
A6	2-methyl-1-butanol	C137326	C_5_H_12_O	0.19 ± 0.0068 ^e^	0.29 ± 0.0401 ^c^	0.14 ± 0.0062 ^f^	0.60 ± 0.0083 ^a^	0.24 ± 0.0023 ^d^	0.41 ± 0.0084 ^b^
A7	(Z)-2-pentenol	C1576950	C_5_H_10_O	0.13 ± 0.0055 ^d^	0.58 ± 0.0721 ^b^	0.72 ± 0.0275 ^a^	0.30 ± 0.0380 ^c^	0.19 ± 0.0019 ^d^	0.12 ± 0.0118 ^d^
A8	1-pentanol	C71410	C_5_H_12_O	0.86 ± 0.0172 ^d^	1.70 ± 0.0471 ^c^	0.48 ± 0.0358 ^e^	2.70 ± 0.0389 ^a^	2.59 ± 0.0216 ^a^	2.06 ± 0.1608 ^b^
A9	3-methyl-1-pentanol	C589355	C_6_H_14_O	0.11 ± 0.0071 ^c^	0.18 ± 0.0213 ^b,c^	0.22 ± 0.0011 ^b^	0.58 ± 0.1136 ^a^	0.22 ± 0.0045 ^b^	0.10 ± 0.0078 ^c^
A10	1-hexanol	C111273	C_6_H_14_O	0.29 ± 0.0272 ^c^	0.58 ± 0.083 ^b^	0.30 ± 0.0452 ^c^	3.92 ± 0.1271 ^a^	0.33 ± 0.0303 ^c^	0.62 ± 0.0304 ^b^
A11	(Z)-3-hexen-1-ol	C928961	C_6_H_12_O	0.06 ± 0.0073 ^b^	0.13 ± 0.0043 ^b^	0.05 ± 0.0009 ^b^	0.07 ± 0.0269 ^b^	0.17 ± 0.0096 ^b^	0.56 ± 0.3009 ^a^
A12	1-octen-3-ol	C3391864	C_8_H_16_O	0.08 ± 0.0026 ^e^	0.21 ± 0.0050 ^b^	0.18 ± 0.0149 ^c^	0.30 ± 0.0165 ^a^	0.10 ± 0.0079 ^d^	0.08 ± 0.0054 ^d,e^
A13	1-octanol	C111875	C_8_H18O	0.20 ± 0.0044 ^b^	0.25 ± 0.0148 ^a^	0.10 ± 0.0125 ^c^	0.26 ± 0.0540 ^a^	0.29 ± 0.0061 ^a^	0.07 ± 0.0055 ^c^
A14	Linalool	C78706	C_10_H_18_O	0.07 ± 0.0032 ^d^	0.47 ± 0.0336 ^c^	3.33 ± 0.3002 ^a^	1.18 ± 0.0309 ^b^	3.57 ± 0.0733 ^a^	0.53 ± 0.0190 ^c^
A15	cis-p-menth-2-en-1-ol	C29803825	C_10_H_18_O	0.08 ± 0.0028 ^c^	0.23 ± 0.0050 ^b^	1.07 ± 0.1186 ^a^	0.34 ± 0.0178 ^b^	1.04 ± 0.0186 ^a^	0.29 ± 0.0070 ^b^
A16	alpha-terpineol	C98555	C_10_H_18_O	0.32 ± 0.0329 ^c^	0.75 ± 0.0402 ^b^	0.61 ± 0.0463 ^b^	0.40 ± 0.0346 ^c^	0.68 ± 0.0337 ^b^	2.69 ± 0.1609 ^a^
Ketones (11)				4.24 ± 0.14 ^e^	9.05 ± 0.29 ^d^	13.1 ± 0.38 ^b^	16.6 ± 0.35 ^a^	13.68 ± 0.47 ^b^	10.28 ± 0.46 ^c^
B1	2-propanone	C67641	C_3_H_6_O	1.02 ± 0.0118 ^f^	1.46 ± 0.0938 ^d^	2.36 ± 0.0673 ^a^	1.93 ± 0.0232 ^c^	2.11 ± 0.0360 ^b^	1.24 ± 0.0584 ^e^
B2	2-butanone	C78933	C_4_H_8_O	2.29 ± 0.0134 ^e^	2.30 ± 0.1082 ^e^	3.33 ± 0.2072 ^d^	7.75 ± 0.1029 ^a^	6.74 ± 0.1524 ^b^	4.43 ± 0.2777 ^c^
B3	2-pentanone	C107879	C_5_H_10_O	0.14 ± 0.0070 ^c^	0.22 ± 0.0060 ^c^	0.62 ± 0.0371 ^b^	1.28 ± 0.0470 ^a^	1.31 ± 0.0982 ^a^	0.14 ± 0.0226 ^c^
B4	3-hydroxy-2-butanone	C513860	C_4_H_8_O_2_	0.09 ± 0.0052 ^e,f^	0.18 ± 0.0136 ^f^	4.31 ± 0.1561 ^a^	0.71 ± 0.1472 ^c^	1.10 ± 0.0550 ^b^	0.37 ± 0.0581 ^d^
B5	4-methyl-3-penten-2-one	C141797	C_6_H_10_O	0.06 ± 0.0036 ^e^	0.43 ± 0.0666 ^a^	0.26 ± 0.0093 ^b,c^	0.12 ± 0.005 ^d,e^	0.18 ± 0.0012 ^c,d^	0.27 ± 0.0659 ^b^
B6	2-hexanone	C591786	C_6_H_12_O	0.08 ± 0.0020 ^f^	0.30 ± 0.0126 ^c^	0.16 ± 0.0058 ^d^	0.12 ± 0.0168 ^e^	0.32 ± 0.0113b ^c^	0.34 ± 0.0244 ^a^
B7	Dihydro-2-methyl-3(2H)furanone	C3188009	C_5_H_8_O_2_	0.03 ± 0.0006 ^d^	0.11 ± 0.0127 ^a^	0.04 ± 0.0034 ^c^	0.05 ± 0.002 ^c,d^	0.07 ± 0.0040 ^b^	0.04 ± 0.0046 ^c,d^
B8	2-heptanone	C110430	C_7_H_14_O	0.05 ± 0.0045 ^c^	0.05 ± 0.0039 ^c^	0.08 ± 0.0106 ^c^	0.80 ± 0.0629 ^a^	0.45 ± 0.0807 ^b^	0.07 ± 0.0156 ^c^
B9	3-octanone	C106683	C_8_H_16_O	0.28 ± 0.1059 ^d^	1.15 ± 0.1785 ^c^	0.99 ± 0.0878 ^c^	3.58 ± 0.2010 ^a^	0.89 ± 0.0387 ^c^	3.06 ± 0.3374 ^b^
B10	6-methyl-5-hepten-2-one	C110930	C_8_H_14_O	0.04 ± 0.0052 ^d^	0.12 ± 0.0061 ^c^	0.51 ± 0.0172 ^a^	0.04 ± 0.004 ^d^	0.25 ± 0.0125 ^b^	0.06 ± 0.0048 ^d^
B11	Coumarin	C91645	C_9_H_6_O_2_	0.16 ± 0.0208 ^b^	2.76 ± 0.3605 ^a^	0.46 ± 0.1660 ^b^	0.24 ± 0.0190 ^b^	0.26 ± 0.0116 ^b^	0.25 ± 0.0331 ^b^
Esters (22)				46.37 ± 0.09 ^a^	44.95 ± 0.69 ^a^	35.62 ± 0.58 ^b^	24.38 ± 0.64 ^c^	32.21 ± 0.65 ^d^	27.53 ± 1.46 ^e^
C1	Methyl acetate	C79209	C_3_H_6_O_2_	0.88 ± 0.0017 ^d^	3.10 ± 0.4606 ^a^	1.37 ± 0.0792 ^c^	2.26 ± 0.0315 ^b^	2.00 ± 0.0138 ^b^	2.05 ± 0.1902 ^b^
C2	Ethyl acetate	C141786	C_4_H_8_O_2_	9.56 ± 0.1527 ^d^	23.09 ± 0.1598 ^a^	19.99 ± 0.276 ^b^	6.22 ± 0.4258 ^e^	12.95 ± 0.3376 ^c^	12.51 ± 0.2807 ^c^
C3	Isopropyl acetate	C108214	C_5_H_10_O_2_	0.03 ± 0.0010 ^b^	0.07 ± 0.0032 ^b^	0.06 ± 0.0050 ^b^	0.09 ± 0.0120 ^b^	0.08 ± 0.0094 ^b^	0.36 ± 0.0702 ^a^
C4	Propyl acetate	C109604	C_5_H_10_O_2_	7.23 ± 0.1196 ^b^	10.54 ± 0.0565 ^a^	1.02 ± 0.0262 ^e^	3.97 ± 0.244 ^d^	3.29 ± 0.0682 ^f^	4.43 ± 0.3286 ^c^
C5	Ethyl 2-methylpropanoate	C97621	C_6_H_12_O_2_	2.15 ± 0.0096 ^b^	1.10 ± 0.1287 ^c^	1.03 ± 0.0846 ^c^	0.49 ± 0.038 ^d^	2.82 ± 0.0523 ^a^	1.10 ± 0.0878 ^c^
C6	2-methylpropyl acetate	C110190	C_6_H_12_O_2_	0.51 ± 0.0207 ^b^	0.16 ± 0.0470 ^c^	0.05 ± 0.0012 ^c^	0.13 ± 0.0045 ^c^	0.06 ± 0.0055 ^c^	0.96 ± 0.2729 ^a^
C7	Methyl 3-methylbutanoate	C556241	C_6_H_12_O_2_	0.28 ± 0.0254 ^b^	0.32 ± 0.0250 ^b^	0.58 ± 0.0571 ^a^	0.09 ± 0.006 ^c,d^	0.15 ± 0.0065 ^c^	0.07 ± 0.0050 ^d^
C8	Ethyl butanoate	C105544	C_6_H_12_O_2_	3.54 ± 0.0363 ^b^	2.72 ± 0.0295 ^d^	4.85 ± 0.0927 ^a^	3.22 ± 0.0397 ^c^	3.28 ± 0.0316 ^c^	0.51 ± 0.0297 ^e^
C9	Ethyl 3-methylbutanoate	C108645	C_7_H_14_O_2_	3.68 ± 0.0484 ^b^	0.38 ± 0.0524 ^f^	2.34 ± 0.0673 ^c^	0.74 ± 0.044 ^d^	4.06 ± 0.0619 ^a^	0.56 ± 0.0662 ^e^
C10	Isoamyl acetate	C123922	C_7_H_14_O_2_	0.61 ± 0.0442 ^b^	0.60 ± 0.2229 ^b^	0.13 ± 0.0069 ^c^	0.65 ± 0.0099 ^c^	0.15 ± 0.0351 ^b^	2.07 ± 0.3881 ^a^
C11	Propyl butanoate	C105668	C_7_H_14_O_2_	1.15 ± 0.0852 ^b^	0.17 ± 0.0107 ^d^	0.96 ± 0.0165 ^c^	1.55 ± 0.0362 ^a^	0.15 ± 0.0082 ^d^	0.11 ± 0.0065 ^d^
C12	Methyl hexanoate	C106707	C_7_H_14_O_2_	0.08 ± 0.0013 ^c^	0.04 ± 0.0000 ^d^	—	0.26 ± 0.0144 ^b^	—	0.37 ± 0.0256 ^a^
C13	Ethyl hexanoate	C123660	C_8_H_16_O_2_	0.69 ± 0.0250 ^c^	0.36 ± 0.0298 ^d^	1.11 ± 0.0056 ^b^	1.93 ± 0.1395 ^a^	0.51 ± 0.0208 ^d^	0.37 ± 0.0930 ^d^
C14	Methyl heptanoate	C106730	C_8_H_16_O_2_	0.11 ± 0.0106 ^a^	0.07 ± 0.0033 ^c,d^	0.06 ± 0.0055 ^d^	0.09 ± 0.002 ^b,c^	0.07 ± 0.0041 ^d^	0.09 ± 0.0097 ^b^
C15	Ethyl 2-hydroxy-4-methylpentanoate-D	C10348477	C_8_H_16_O_3_	0.07 ± 0.0084 ^d^	0.11 ± 0.0094 ^b,c^	0.14 ± 0.0022 ^b^	0.11 ± 0.017 ^b,c^	0.24 ± 0.0216 ^a^	0.10 ± 0.0060 ^c^
C16	Ethyl 2-hydroxy-4-methylpentanoate-M	C10348477	C_8_H_16_O_3_	0.15 ± 0.0465 ^d^	0.21 ± 0.0244 ^c,d^	0.13 ± 0.0103 ^d^	0.30 ± 0.0075 ^c^	0.80 ± 0.0183 ^a^	0.59 ± 0.0908 ^b^
C17	Ethyl 3-hydroxyhexanoate-M	C2305251	C_8_H_16_O_3_	4.06 ± 0.0775 ^a^	0.71 ± 0.1820 ^b^	0.48 ± 0.0454 ^c^	0.35 ± 0.0308 ^c^	0.38 ± 0.0354 ^c^	0.33 ± 0.0180 ^c^
C18	Ethyl 3-hydroxyhexanoate-D	C2305251	C_8_H_16_O_3_	10.59 ± 0.382 ^a^	0.46 ± 0.0281 ^b^	0.48 ± 0.0129 ^b^	0.42 ± 0.0286 ^b^	0.45 ± 0.0378 ^b^	0.37 ± 0.0105 ^b^
C19	Ethyl heptanoate-D	C106309	C_9_H_18_O_2_	0.22 ± 0.0113 ^a^	0.09 ± 0.0048 ^c,d^	0.10 ± 0.0042 ^b^	0.09 ± 0.007 ^c,d^	0.10 ± 0.0036 ^c^	0.08 ± 0.0066 ^d^
C20	Ethyl heptanoate-M	C106309	C_9_H_18_O_2_	0.41 ± 0.0751 ^a^	0.16 ± 0.0633 ^c^	0.14 ± 0.0106 ^c^	0.30 ± 0.0205 ^b^	0.15 ± 0.0201 ^c^	0.09 ± 0.0110 ^c^
C21	(Z)-3-hexenyl butanoate	C16491364	C_10_H_18_O_2_	0.17 ± 0.0059 ^b,c^	0.19 ± 0.0232 ^a,b^	0.17 ± 0.0099 ^b,c^	0.21 ± 0.0127 ^a^	0.18 ± 0.0227 ^a,b^	0.14 ± 0.0044 ^c^
C22	Geranyl acetate	C105873	C_12_H_20_O_2_	0.23 ± 0.0248 ^d^	0.33 ± 0.0256 ^c^	0.44 ± 0.0350 ^b^	0.94 ± 0.0546 ^a^	0.34 ± 0.0289 ^c^	0.29 ± 0.0071 ^c,d^
Aldehydes (25)				16.16 ± 0.11 ^c^	15.78 ± 0.65 ^c^	18.79 ± 0.76 ^a^	17.55 ± 0.69 ^b^	19.13 ± 0.16 ^a^	10 ± 0.04 ^d^
D1	Propanal	C123386	C_3_H_6_O	0.82 ± 0.0626 ^f^	2.47 ± 0.1159 ^c^	5.36 ± 0.1381 ^a^	2.21 ± 0.011 ^d^	4.03 ± 0.0787 ^b^	1.25 ± 0.0483 ^e^
D2	Butanal	C123728	C_4_H_8_O	2.00 ± 0.0091 ^c^	1.25 ± 0.0391 ^d^	1.99 ± 0.1079 ^c^	3.67 ± 0.0515 ^a^	3.78 ± 0.0348 ^a^	2.75 ± 0.2000 ^b^
D3	3-methylbutanal	C590863	C_5_H_10_O	0.15 ± 0.0067 ^c^	0.84 ± 0.0302 ^b^	0.21 ± 0.0166 ^c^	0.96 ± 0.049 ^a,b^	0.86 ± 0.0406 ^a,b^	0.98 ± 0.1069 ^a^
D4	2-methylbutanal	C96173	C_5_H_10_O	0.09 ± 0.0026 ^d^	0.40 ± 0.0081 ^c^	1.21 ± 0.0397 ^a^	0.67 ± 0.0614 ^b^	0.70 ± 0.0233 ^b^	0.38 ± 0.0114 ^c^
D5	Pentanal	C110623	C_5_H_10_O	0.25 ± 0.0066 ^d^	0.32 ± 0.0087 ^c^	1.32 ± 0.0241 ^a^	0.28 ± 0.072 ^c,d^	0.42 ± 0.0124 ^b^	0.13 ± 0.0129 ^e^
D6	(E)-2-pentenal	C1576870	C_5_H_8_O	0.07 ± 0.0026 ^b^	0.15 ± 0.0053 ^b^	0.16 ± 0.0242 ^b^	0.67 ± 0.1905 ^a^	0.06 ± 0.0013 ^b^	0.03 ± 0.0005 ^b^
D7	3-methyl-2-butenal	C107868	C_5_H_8_O	0.11 ± 0.0099 ^b,c^	0.20 ± 0.0185 ^a^	0.19 ± 0.0149 ^a^	0.07 ± 0.0138 ^c^	0.19 ± 0.0109 ^a^	0.12 ± 0.0354 ^b^
D8	Hexanal-M	C66251	C_6_H_12_O	1.31 ± 0.1736 ^b^	1.94 ± 0.1741 ^a^	1.34 ± 0.1765 ^b^	1.10 ± 0.0302 ^b^	0.66 ± 0.0293 ^c^	0.30 ± 0.0246 ^d^
D9	Hexanal-D	C66251	C_6_H_12_O	3.50 ± 0.0904 ^b^	1.61 ± 0.0508 ^e^	3.18 ± 0.1823 ^c^	2.07 ± 0.102 ^d^	5.13 ± 0.1578 ^a^	0.69 ± 0.0736 ^f^
D10	Methional	C3268493	C_4_H_8_OS	0.05 ± 0.0536 ^d^	0.16 ± 0.1638 ^b^	0.24 ± 0.2373 ^a^	—	0.11 ± 0.1112 ^c^	0.06 ± 0.0037 ^d^
D11	Heptanal	C111717	C_7_H_14_O	0.81 ± 0.1290 ^a^	0.43 ± 0.0297 ^b^	0.21 ± 0.0168 ^c^	0.25 ± 0.0164 ^c^	0.30 ± 0.0096 ^c^	0.18 ± 0.0095 ^c^
D12	(Z)-4-heptenal-D	C6728310	C_7_H_12_O	2.47 ± 0.0436 ^a^	0.07 ± 0.0048 ^d^	0.07 ± 0.0092 ^d^	0.40 ± 0.0130 ^b^	0.07 ± 0.0099 ^d^	0.18 ± 0.0114 ^c^
D13	(Z)-4-heptenal-M	C6728310	C_7_H_12_O	0.80 ± 0.0167 ^b^	0.57 ± 0.0944 ^c^	0.98 ± 0.0551 ^a^	0.28 ± 0.090 ^d^	0.24 ± 0.0264 ^d^	0.39 ± 0.0540 ^d^
D14	(E)-2-heptenal-D	C18829555	C_7_H_12_O	0.04 ± 0.0036 ^b^	0.29 ± 0.0331 ^b^	0.04 ± 0.0000 ^b^	1.20 ± 0.4321 ^a^	0.04 ± 0.0000 ^b^	0.05 ± 0.0048 ^b^
D15	(E)-2-heptenal-M	C18829555	C_7_H_12_O	0.12 ± 0.0079 ^d^	0.56 ± 0.0612 ^b^	0.11 ± 0.0222 ^d^	0.49 ± 0.0395 ^b^	0.21 ± 0.0017 ^c^	0.71 ± 0.0382 ^a^
D16	Benzaldehyde	C100527	C_7_H_6_O	0.03 ± 0.0057 ^d^	0.08 ± 0.0083 ^b^	0.05 ± 0.0000 ^c,d^	0.05 ± 0.0045 ^c^	0.15 ± 0.0154 ^a^	0.05 ± 0.0026 ^c,d^
D17	2,4-heptadienal	C5910850	C_7_H_10_O	0.29 ± 0.0824 ^c^	1.50 ± 0.1896 ^a^	0.73 ± 0.0547 ^b^	0.37 ± 0.0155 ^c^	0.28 ± 0.0199 ^c^	0.18 ± 0.0164 ^c^
D18	Octanal	C124130	C_8_H_16_O	0.55 ± 0.0617 ^c^	1.10 ± 0.0750 ^a^	0.15 ± 0.0076 ^d^	0.71 ± 0.0154 ^b^	0.22 ± 0.0048 ^d^	0.48 ± 0.0188 ^c^
D19	Benzeneacetaldehyde	C122781	C_8_H_8_O	0.11 ± 0.0027 ^d^	0.69 ± 0.0465 ^a^	0.14 ± 0.0151 ^c,d^	0.40 ± 0.0358 ^b^	0.40 ± 0.0174 ^b^	0.18 ± 0.0178 ^c^
D20	Nonanal	C124196	C_9_H_18_O	0.15 ± 0.0079 ^b^	0.13 ± 0.0024 ^b,c^	0.1 ± 0.0070 ^d^	0.30 ± 0.0324 ^a^	0.08 ± 0.0059 ^d^	0.11 ± 0.0067 ^c,d^
D21	(Z)-4-decenal-D	C21662099	C_10_H_18_O	0.15 ± 0.0370 ^a^	0.07 ± 0.0080 ^b^	0.07 ± 0.0023 ^b^	0.05 ± 0.0011 ^b^	0.06 ± 0.0074 ^b^	0.06 ± 0.0038 ^b^
D22	(Z)-4-decenal-M	C21662099	C_10_H_18_O	1.39 ± 0.1362 ^a^	0.31 ± 0.0131 ^c,d^	0.28 ± 0.0506 ^c,d^	0.62 ± 0.0394 ^b^	0.35 ± 0.0241 ^c^	0.19 ± 0.0083 ^d^
D23	2-decenal	C3913711	C_10_H_18_O	0.40 ± 0.0201 ^a^	0.24 ± 0.0291 ^c,d^	0.34 ± 0.0028 ^b^	0.28 ± 0.0077 ^c^	0.40 ± 0.0174 ^a^	0.22 ± 0.0137 ^d^
D24	(E,E)-2,4-nonadienal	C5910872	C_9_H_14_O	0.30 ± 0.0269 ^a^	0.13 ± 0.0119 ^c^	0.11 ± 0.0093 ^c^	0.17 ± 0.0051 ^b^	0.11 ± 0.0072 ^c^	0.12 ± 0.0072 ^c^
D25	(E,E)-2,4-decadienal	C25152845	C_10_H_16_O	0.19 ± 0.0210 ^a^	0.26 ± 0.0257 ^a^	0.24 ± 0.0150 ^a^	0.25 ± 0.0267 ^a^	0.26 ± 0.0352 ^a^	0.22 ± 0.0420 ^a^
Acids (2)				0.16 ± 0.01 ^c^	0.26 ± 0.01 ^b^	0.51 ± 0.02 ^a^	0.09 ± 0.01 ^d^	0.17 ± 0.01 ^c^	0.09 ± 0.01 ^d^
E1	Propanoic acid	C79094	C_3_H_6_O_2_	—	—	0.19 ± 0.0163 ^a^	0.04 ± 0.0018 ^b^	0.09 ± 0.0043 ^c^	—
E2	3-methylbutanoic acid	C503742	C_5_H_10_O_2_	0.16 ± 0.0063 ^c^	0.26 ± 0.0098 ^b^	0.32 ± 0.0071 ^a^	0.05 ± 0.0010 ^e^	0.08 ± 0.0037 ^d^	0.09 ± 0.0096 ^d^
Alkenes (11)				17.38 ± 0.09 ^a^	4.79 ± 0.23 ^c^	6.61 ± 0.47 ^b^	5.31 ± 0.17 ^c^	5.44 ± 0.17 ^c^	17.44 ± 0.44 ^a^
F1	Tricyclene	C508327	C_10_H_16_	0.08 ± 0.0029 ^e^	0.21 ± 0.0399 ^c^	0.27 ± 0.0139 ^b^	0.07 ± 0.0091 ^e^	0.12 ± 0.0047 ^d^	0.69 ± 0.0070 ^a^
F2	Alpha-pinene	C80568	C_10_H_16_	0.13 ± 0.0052 ^d^	0.29 ± 0.0284 ^b^	0.46 ± 0.0332 ^a^	0.10 ± 0.009 ^d^	0.21 ± 0.0213 ^c^	0.27 ± 0.0218 ^b^
F3	Beta-pinene-M	C127913	C_10_H_16_	0.30 ± 0.0090 ^d^	0.98 ± 0.0938 ^c^	2.04 ± 0.2142 ^b^	1.23 ± 0.1524 ^c^	1.89 ± 0.0840 ^b^	3.57 ± 0.0228 ^a^
F4	Beta-pinene-D	C127913	C_10_H_16_	0.18 ± 0.0091 ^d^	0.27 ± 0.0058 ^d^	0.42 ± 0.0375 ^c^	0.93 ± 0.0477 ^b^	0.51 ± 0.0340 ^c^	4.90 ± 0.1473 ^a^
F5	Beta-pinene-T	C127913	C_10_H_16_	0.05 ± 0.0114 ^b^	0.07 ± 0.0043 ^b^	0.09 ± 0.0153 ^b^	0.08 ± 0.0116 ^b^	0.08 ± 0.0047 ^b^	1.40 ± 0.0648 ^a^
F6	Myrcene	C123353	C_10_H_16_	0.15 ± 0.0159 ^c^	0.20 ± 0.0038 ^c^	0.31 ± 0.0129 ^b^	0.29 ± 0.0596 ^b^	0.23 ± 0.0078 ^b,c^	0.58 ± 0.0646 ^a^
F7	Alpha-phellandrene-M	C99832	C_10_H_16_	2.33 ± 0.0553 ^a^	1.44 ± 0.2083 ^b^	1.54 ± 0.1210 ^b^	1.07 ± 0.1216 ^c^	1.39 ± 0.0468 ^b^	2.40 ± 0.1704 ^a^
F8	Alpha-phellandrene-D	C99832	C_10_H_16_	13.61 ± 0.040 ^a^	0.45 ± 0.1576 ^c^	0.37 ± 0.0304 ^c^	1.06 ± 0.1179 ^b^	0.29 ± 0.0135 ^c^	0.39 ± 0.0560 ^c^
F9	Beta-ocimene	C13877913	C_10_H_16_	0.27 ± 0.0141 ^c^	0.46 ± 0.0313 ^b^	0.44 ± 0.0324 ^b^	0.19 ± 0.019 ^d^	0.33 ± 0.0126 ^c^	1.85 ± 0.0619 ^a^
F10	Gamma-terpinene	C99854	C_10_H_16_	0.14 ± 0.0053 ^c^	0.17 ± 0.0235 ^c^	0.34 ± 0.0266 ^b^	0.17 ± 0.0107 ^c^	0.16 ± 0.0248 ^c^	0.72 ± 0.0114 ^a^
F11	Terpinolene	C586629	C_10_H_16_	0.14 ± 0.0108 ^d^	0.24 ± 0.0081 ^c^	0.32 ± 0.0206 ^b^	0.12 ± 0.014 ^d^	0.22 ± 0.0178 ^c^	0.67 ± 0.0404 ^a^
Others (8)				3.87 ± 0.02 ^e^	7.18 ± 0.27 ^b^	5.47 ± 0.15 ^d^	6.68 ± 0.01 ^c^	4.28 ± 0.05 ^e^	18.11 ± 0.42 ^a^
G1	Dimethylamine	C124403	C_2_H_7_N	2.11 ± 0.2938 ^b^	2.70 ± 0.1530 ^a^	2.19 ± 0.1176 ^b^	1.72 ± 0.0640 ^c^	1.36 ± 0.0208 ^d^	0.43 ± 0.0143 ^e^
G2	Dimethyl disulfide	C624920	C_2_H_6_S_2_	0.06 ± 0.0025 ^c^	0.04 ± 0.0001 ^c^	1.60 ± 0.0556 ^a^	—	0.34 ± 0.0119 ^b^	—
G3	2-acetylfuran	C1192627	C_6_H_6_O_2_	0.04 ± 0.0048 ^c^	0.05 ± 0.0073 ^c^	0.04 ± 0.0001 ^c^	0.18 ± 0.0128 ^a^	0.08 ± 0.0036 ^b^	0.07 ± 0.0137 ^b^
G4	2-pentylfuran	C3777693	C_9_H_14_O	0.10 ± 0.0195 ^d^	0.35 ± 0.0210 ^b^	0.19 ± 0.0133 ^c^	0.55 ± 0.0777 ^a^	0.22 ± 0.0032 ^c^	0.27 ± 0.0493 ^b,c^
G5	1,8-cineole-D	C470826	C_10_H_18_O	0.48 ± 0.0311 ^b^	0.35 ± 0.0242 ^b,c^	0.15 ± 0.0072 ^c^	0.47 ± 0.0178 ^b^	0.19 ± 0.0176 ^c^	8.57 ± 0.2251 ^a^
G6	1,8-cineole-M	C470826	C_10_H_18_O	0.76 ± 0.2324 ^d^	3.21 ± 0.2589 ^b^	1.02 ± 0.0743 ^d^	3.54 ± 0.1134 ^b^	1.69 ± 0.0605 ^c^	8.49 ± 0.2093 ^a^
G7	2,3-diethyl-5-methylpyrazine	C18138040	C_9_H_14_N_2_	0.23 ± 0.0039 ^a,b^	0.20 ± 0.0477 ^b^	0.14 ± 0.0264 ^c^	0.10 ± 0.0103 ^c^	0.28 ± 0.0186 ^a^	0.09 ± 0.0053 ^c^
G8	Methyl chavicol	C140670	C_10_H_12_O	0.08 ± 0.0036 ^d^	0.28 ± 0.0330 ^a^	0.14 ± 0.0193 ^b,c^	0.10 ± 0.010 ^c,d^	0.13 ± 0.0188 ^c^	0.18 ± 0.0162 ^b^

The -m and -d following some substances in the list indicate Monomer and Dimer of the same substance. Different letters in the same row represent significant differences (*p* < 0.05).

## Data Availability

The raw data have been submitted to the NCBI Sequence Read Archive (SRA), with the accession number PRJNA860537.

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
