# Peer review of "Correlation and Difference between Core Micro-Organisms and Volatile Compounds of Suan Rou from Six Regions of China"

_foods, 2022, doi:10.3390/foods11172708_

Round 1

Reviewer 1 Report

Line(s)          Comment

72-83           It is not clear if the samples were from producers in the respective areas or if the samples were made by the scientists with the respective ingredients in Table 1.

150-164       These sentences are instructions for manuscript preparation and their inclusion reflects a deficiency of the authors in reading the paper before it was submitted.

189               Use of “significant” and the probability level together are redundant.

310-311       Reference(s) needed to validate this statement as factual.

340-341       Reference(s) needed to validate this statement as factual.

484-503       These sentences repeat the results and are not conclusions based on the results that indicate how and by whom the results can be used.

528-630       There are discrepancies in the citation format among the journal article titles.

Author Response

Dear Reviewer 1:

    We would like to thank you for your careful and thorough reading of this manuscript and for the thoughtful comments and constructive suggestions. Please see the attachment.

Reviewer 2 Report

The article is interesting, but it does contain some inaccuracies. They concern:

title: should be a bit shorter, in its current form it is quite illegible.

This is also related to the suggestion for keywords that are too many and are the repetition of the title. In addition, these should be phrases, the most important keys that the reader can use to find the article in search engines. The authors are requested to correct these inaccuracies.

Abstract:

There is no information in this section of the article about what the samples were and exactly how many different regions of China they came from. Authors should provide the characteristics of the samples that have been analyzed. It is not known how? There is no specific summary of the overall research that is the subject of this article.

Introduction: It is quite laconic and contains basic information, generally rather popular science. There are no significant issues that make the reader feel meaningful in the experiment performed with respect to the relevant research on the scope of the topic.

The authors did not indicate the novelty of their experiment. How does the currently presented research differ from the previous ones that have already been published? Therefore, please outline the background around the analyzed raw material, especially since the authors later provide detailed information on the characteristics and occurrence as well as their possible applications.

However, there is no information that the research was aimed at a comparative analysis of different fermentation processes at different times and at different temperatures, which in turn influenced the antioxidant activity of tea infusions. There is also no information about what substances were marked - this applies to the main volatile compounds. The authors only provided the groups to which they belonged.

Chapter: 2. Materials and Methods

Subsection: 2.1. Sample collection

There is no information about the fermentation conditions. This should be completed.

Subsection: 2.4. Analysis of metabolites

This chapter should be called: 2.4. Statistical analysis.

Author Response

Dear Reviewer 2:

    We would like to thank you for your careful and thorough reading of this manuscript and for the thoughtful comments and constructive suggestions. Please see the attachment.

Reviewer 3 Report

The current study on analysis of correlation and difference between core microorganisms and volatile organic compounds of suanrou from different regions of China is interesting and has great local significance.  

Line 28: what do you mean by Suan rou? Do you mean Suanrou? If yes, you have already abbreviated it on first seen and please use SR

Line 43: please don’t start sentence with abbreviation

Line 42: Do you mean volatile organic compounds? Its organic or flavor compound

Line 56: volatile organic compounds? Isn’t VOCs, if yes please use VOCs instead of volatile organic compound

Line 67: you are giving two abbreviation that are HTS and GC-IMS. Do you mean high-throughput sequencing and gas chromatographyion mobility spectrometry? If yes, please terms should be abbreviated on first seen both in abstract and introduction and used in subsequent sections. Thanks

Line 67-68: in sentence ‘The core differential microorganisms and 67 differential VOCs were determined by O2PLS-DA analysis………..’ O2PLS-DA analysis is used. What do you mean by O2PLS-DA analysis? do you mean bidirectional orthogonal projections to latent structures.. if yes please give full term and abbreviate it

Line 84: actually you collected samples from six sites in triplicates? You may clearly mention it in the abstract as well as the materials and methods section. Don’t you think sample size and replicates are  too small to verify the results

Line 121: LDA term must be abbreviated on first seen. You did not use LDA frequently, so I will suggest not to abbreviate term

Line 137: ‘Analysis of metabolites’ is confusing with statistical analysis

Line 144: O2PLS already suggested to abbreviate

Line 149: ‘All experiments were performed in triplicate, and the data were ex- 148 pressed as means ± standard deviations (SD)’ remove SD from sentence

Line 150-164: what is this? ‘The Materials and Methods should be described with sufficient details to allow others to replicate and build on the published results. Please note that the publication of your  manuscript implicates that you must make all materials, data, computer code, and protocols associated with the publication available to readers. Please disclose at the submission stage any restrictions on the availability of materials or information. New methods and protocols should be described in detail while well-established methods can be briefly described and appropriately cited. Research manuscripts reporting large datasets that are deposited in a publicly available database should specify where the data have been deposited and provide the relevant accession numbers. If the accession numbers have not yet been obtained at the time of submission, please state that they will be provided during review. They must be provided prior to publication. Interventionary studies involving animals or humans, and other studies that require ethical approval, must list the authority that provided approval and the corresponding ethical approval code’

Line 175-176: ‘The higher the Shannon index, the lower the Simpson index and  the higher the Alpha diversity of the microbial community in the SR sample’ this sentence is not clear. Please rewrite it

Line 200: italic Firmicutes, not only for it but for other species  and consider it throughout the manuscript

Line 203-204: would you like to explain what was the work of by Lv [12] and Wang [24] in the sentence  ‘The results were 203 consistent with the results reported by Lv [12] and Wang [24]’

Line 223: what do you mean by ‘two bacteria’ do you mean ‘Staphylococcus and Pediococcus’ if yes did you correlate it in the subsequent section?

Line 271: volatile organic compounds? Isn’t VOCs, if yes please use VOCs instead of volatile organic compound

May I know the reason why you did not check statistical different for flavor compounds between treatment. Although you provided SD of each location, but you did not mention the difference between location. If I look at

table 1

HNXX Pork belly, Rice, Pepper, Salt

GXRS Pork belly, Chili, Salt

GZZY Pork belly, Rice, Salt

GZLB Pork belly, Millet, Pepper, Salt

GZCJ Pork belly, Millet, Chili, Pepper, Salt

GZJP Pork belly, Chili, Pepper, Salt

So accessories may influence these flavor compound but you did not take it in consideration. How can you justify this?

Conclusion is too long. Please short it

Author Response

Dear Reviewer 3:

    We would like to thank you for your careful and thorough reading of this manuscript and for the thoughtful comments and constructive suggestions. Please see the attachment.
